# Dapansutrile Regulates Mitochondrial Oxidative Stress and Reduces Hepatic Lipid Accumulation in Diabetic Mice

**DOI:** 10.3390/cimb47030148

**Published:** 2025-02-25

**Authors:** Ying Wu, Jiaqiang Zhou

**Affiliations:** Department of Endocrinology and Metabolism, Sir Run Run Shaw Hospital, Zhejiang University School of Medicine, Hangzhou 310016, China; 12018415@zju.edu.cn

**Keywords:** Dapansutrile, mitochondria, NLRP3, hepatic lipid accumulation

## Abstract

(1) Background: Hepatic lipid accumulation is the initial factor in metabolic-associated fatty liver disease (MAFLD) in type 2 diabetics, leading to accelerated liver damage. The NOD-like receptor protein 3 (NLRP3) inflammasome plays a critical role in this process. Dapansutrile (DAPA) is a novel NLRP3 inflammasome inhibitor; however, its effect on ectopic lipid accumulation in the liver remains unclear. This study aimed to investigate the therapeutic effect of DAPA on hepatic lipid accumulation in a diabetic mouse model and its potential mechanisms. (2) Methods: The effects of DAPA on hepatic ectopic lipid deposition and liver function under metabolic stress were evaluated in vivo using db/db and high-fat diet (HFD) + streptozotocin (STZ) mouse models. Additionally, the role and mechanism of DAPA in cellular lipid deposition, mitochondrial oxidative stress, and inflammation were assessed in HepG2 cells treated with free fatty acids (FFA) and DAPA. (3) Results: Our findings indicated that DAPA treatment improved glucose and lipid metabolism in diabetic mice, particularly addressing liver heterotopic lipid deposition and insulin resistance. DAPA treatment also ameliorated lipid accumulation and mitochondrial-related functions and inflammation in HepG2 cells through the NLRP3-Caspase-1 signaling axis. (4) Conclusions: Targeting NLRP3 with DAPA may represent a novel therapeutic approach for diabetes-related fatty liver diseases.

## 1. Introduction

The prevalence of metabolic-associated fatty liver disease (MAFLD) among patients with type 2 diabetes (T2DM) is rising, yet this comorbidity often receives insufficient clinical attention [1]. In recent years, with the concept of metabolic dysfunction-associated fatty liver disease (MAFLD) proposed, T2DM-associated MAFLD has received increasing attention from scholars [2]. The lack of a good treatment strategy is an urgent problem to be solved for T2DM-related MAFLD, which has significant clinical significance. In addition, overweight and obesity frequently result in ectopic fat deposition, with hepatic lipid accumulation being the most prevalent form. Excessive lipid deposition in the liver not only leads to MAFLD but also significantly increases the risk of T2DM, cardiovascular disease, and other metabolic disorders. While the specific factors contributing to the risk of MAFLD and T2DM are not fully understood, liver lipid accumulation has been linked to liver insulin resistance and inflammation. These are two hallmark features of MAFLD. In the early stages of liver disease, the accumulation of hepatic lipid metabolites, mitochondrial oxidative stress, and impaired lipoprotein secretion may contribute to the risk of developing T2DM. Conversely, in the advanced stages of liver disease, liver inflammation and fibrosis become more prominent contributors to disease progression. Chronic low-grade inflammation and mitochondrial dysfunction exacerbate liver damage, driving the progression of MAFLD [3].

Liver inflammation and mitochondrial homeostasis disruption play pivotal roles in the onset and progression of hepatic lipid accumulation [4]. Exposure to high levels of fat can lead to the formation of harmful lipid intermediates, such as diacylglycerols and ceramides, which can impair the function of the endoplasmic reticulum and mitochondria. The accumulation of dysfunctional mitochondria can increase the production of reactive oxygen species (ROS), triggering downstream inflammatory responses. This cascade damages liver cells by disrupting energy supply and perpetuating a pathological cycle, ultimately resulting in liver dysfunction [5].

The interplay between mitochondrial dysfunction and inflammatory responses is a critical factor in this context. This relationship is particularly evident in the role of inflammasomes, which serve as key mediators of the inflammatory process. Inflammasomes can be activated by danger signals such as mitochondrial ROS. The NOD-like receptor family pyridine domain 3 (NLRP3) inflammasome is currently the most extensively studied inflammasome and widely expressed in liver cells. The activation of inflammasomes by NLRP3 promotes the progression of hepatic lipid accumulation [6]. The NLRP3 inflammasome recruits ASC and Caspase-1 to promote the release of IL-1β and IL-18, leading to cell pyroptosis and severe inflammation [7]. Gasdermin D (GSDMD) is the executor and key regulator of cell pyroptosis [8]. Mitochondrial oxidative stress in liver cells leads to the release of mitochondrial ROS, which activates NLRP3 and subsequently impairs liver cell function.

Inflammasome inhibitors are an emerging research field. MCC950, a NLRP3 inhibitor, has shown potential to alleviate the progression of non-alcoholic steatohepatitis. However, due to its significant hepatotoxicity, its use is limited, and it is not suitable for clinical treatment [9]. Other NLRP3 inflammasome inhibitors such as VENT-02 are mostly in phase I clinical trials, and their safety is unclear. So far, the US Food and Drug Administration has not approved any drugs specifically targeting NLRP3 for clinical use. Dapansutrile (DAPA) is a new specific NLRP3 inflammasome inhibitor that has been clinically studied for various diseases, especially gout. DAPA has entered phase II–III clinical trials with positive results and good safety [10,11]. However, there have been no reports on DAPA in the field of T2DM associated MAFLD. The combination of T2DM and MAFLD has become a new challenge in the fields of liver disease and metabolism in China.

Some marketed drugs have been shown to be effective in metabolic-related fatty liver disease in clinical trials, such as pioglitazone, GLP-1RA, SGLT-2 inhibitors, vitamin E, and ketoconazole, but these drugs have varying degrees of drug side effects. Pioglitazone can improve liver histology, but there are risks of edema, weight gain, bladder cancer, and bone density reduction, so its clinical application is limited. GLP-1RA improves liver histology but has gastrointestinal adverse reactions. SGLT-2 inhibitors can reduce fat content and improve liver histology, but their effects on liver fibrosis still need further research. Vitamin E can effectively improve liver histology but may increase the risk of prostate cancer. Ketone theobromine can improve lobular inflammation and NAS scores, but cannot improve steatosis, ballooning, and fibrosis. Currently, Rezdiffra is the only drug approved by the FDA for the treatment of MAFLD. Rezdiffra is effective in T2DM (with or without MAFLD development), although real-world data is inferior to randomized clinical trial studies. Overall, Rezdiffra may be beneficial for T2DM patients [12]. Rezdiffra improves mitochondrial function by regulating genes involved in mitochondrial biogenesis/autophagy, fatty acid beta oxidation, and lipid phagocytosis, reducing hepatic triglyceride levels and thus lowering lipid toxicity [13]. DAPA has similar effects to Rezdiffra in improving mitochondrial function and reducing lipotoxicity mechanisms. In addition, DAPA can target NLRP3, reduce inflammatory factors and responses, and explore the possibility of improving MAFLD from an inflammatory perspective. However, several medications already on the market have shown efficacy in clinical trials for MAFLD. These include pioglitazone, GLP-1 receptor agonists (GLP-1RA), SGLT-2 inhibitors, vitamin E, and pentoxifylline. Targeting mitochondria and regulating their function is a promising new approach for the development of drugs to treat MAFLD. New potential therapeutic drugs such as the ketogenic diet, melatonin, liraglutide, injection of healthy mitochondria, and mitochondrial-targeted redox nanoparticles are worth studying in clinical trials. Nonetheless, these drugs are currently facing safety or side effect issues and challenges. Therefore, due to its good safety, DAPA has entered phase II–III clinical trials and has unique advantages in targeting the inflammatory and antioxidant stress mechanisms of NLRP3. Strategies aimed at improving hepatic lipid accumulation in MAFLD may help reduce the risk of T2DM by improving insulin resistance, mitochondrial oxidative stress, and chronic inflammation. In this study, we used two diabetes mouse models and HepG2 cells both in vivo and in vitro to investigate the effects of DAPA on liver cell function under metabolic stress and its potential molecular mechanisms.

## 2. Materials and Methods

### 2.1. Animal Models

Male C57BL/6 mice, db/db mice, and db/m mice (5 weeks old) were purchased from Gem Pharmatech Co., Ltd. (Nanjing, China). We developed two diabetic mice models. Both animal procedures were conducted in accordance with the approved protocol by the Institutional Animal Care and Use Committee of Zhejiang University and adhered to the Institutional Laboratory Animal Care and Guidelines (ethical approval number: ZJU20230454, date: 21 November 2023).

The HFD + STZ mouse model mice were randomly assigned to the control group (CTR, normal diet) or the HFD group (60% kcal% fat, Research Diets, D12492). After 30 days of feeding, 30 mg/kg of STZ (freshly dissolved in 0.1 mol/L sodium citrate buffer, pH = 4.2) was administered intraperitoneally to the HFD mice for one day, while the control group was administered the same volume of physiological saline. Blood glucose levels ≥ 11.1 mmol/L twice (approximately 12 h apart) were considered successful models.

Fasting blood glucose levels in db/db mice were monitored weekly until 8 weeks of age. We used a db/db mouse model whose fasting blood glucose levels were measured ≥ 11.1 mmol/L twice (approximately 12 h apart). The model mice were divided into two groups: the model group (treated with an intraperitoneal injection of physiological saline) and the DAPA group, which received a daily intraperitoneal injection of DAPA at a dose of 200 mg/kg (dissolved in physiological saline) [14]. After 8 weeks of DAPA treatment in the HFD + STZ mouse model and db/db mice, the mice were sacrificed, and their tissues were harvested.

### 2.2. Oral Glucose Tolerance Test (OGTT) and Insulin Tolerance Test (ITT) Assays

GTTs and ITTs were performed as previously described [15]. For the GTT, the db/db or the HFD + STZ model mice were fasted overnight (approximately 16 h). Tail-vein blood samples were collected at 0, 15, 30, 60, and 120 min after administering 1 g/kg body weight of oral glucose. A glucometer (ACCU, Roche, Basel, Switzerland) was used to measure the blood glucose levels from 0 to 120 min following the glucose challenge to determine the AUC. For the ITT, mice were fasted for 4 h and intraperitoneally injected with an insulin saline solution (1 U/kg body weight). Blood glucose levels were measured via tail-snip blood sampling at 0, 15, 30, 60, and 120 min after injection. Mouse blood samples were collected for insulin concentration measurement using a mouse insulin immunoassay kit (MS300, Ezassay, Shenzhen, China) following the manufacturer’s instructions.

### 2.3. Hematoxylin and Eosin (H&E) Staining and Oil Red O Staining Analysis

At the conclusion of the study, mice were euthanized, and livers from the different groups were immediately excised and thoroughly rinsed with phosphate-buffered saline (PBS). Liver tissue sections were fixed in 4% paraformaldehyde, embedded in paraffin, sectioned at 5 μm thickness, and mounted onto slides. Hematoxylin and eosin (H&E) staining was performed on the sections to evaluate liver histology [16]. For Oil Red O staining, frozen liver tissues were embedded and sectioned at 5 μm thickness. These sections were stained using an Oil Red O kit (Nanjing Biotechnology Institute, Nanjing, China). Images were captured under a microscope (Carl Zeiss, Jena, Germany).

### 2.4. Serum and Liver Biochemical Parameters

All samples underwent pre-processing according to the manufacturer’s protocol. In brief, an appropriate amount of liver tissue was taken, and 9 times the volume of homogenate medium was added before mechanically homogenizing under ice water bath conditions at 2500 rpm, centrifuging for 10 min, and taking the supernatant for testing. Using a commercial testing kit (Nanjing Jiancheng Biotechnology Research Institute, Nanjing, China), serum alanine aminotransferase (ALT), aspartate aminotransferase (AST), liver triglycerides (TG), and total cholesterol (TC) levels were measured according to the manufacturer’s protocol. After carefully adding the corresponding reagents and samples to the blank, standard, and sample wells, the microplate was gently shaken to ensure thorough mixing. Following this, the microplate was incubated at room temperature for the duration specified in the protocol. Once the incubation period was complete, the absorbance of each well was measured using a microplate reader. Finally, calculate the results according to the formula provided in the kit instructions. During the assays, standard curves were generated by plotting the absorbance values against the gradient concentrations of the standards provided with the kits.

### 2.5. Cell Culture and Therapy

HepG2, a human hepatocarcinoma cell line obtained from ATCC, was cultured in DMEM medium (Invitrogen, Carlsbad, CA, USA) supplemented with 10% fetal bovine serum (FBS) and 1× GlutaMax (Invitrogen). Cells were seeded in a 6-well plate at a density of 4 × 10^5^ cells per well. After seeding, the cells were treated with either 0.5 mM free fatty acids (FFA) alone [17] or 0.5 mM FFA combined with 10 μM DAPA for 24 h [18]. The experimental groupings were as follows: CTR group: PBS (control); FFA group: 0.5 mM FFA; DAPA group: 0.5 mM FFA + 10 μM DAPA.

Preparation of FFA Mother Liquor:Palmitic Acid (PA): (1) Dissolve 21.18 mg of palmitic acid in 0.4 mL of anhydrous ethanol. (2) Combine 0.4 mL of the above solution with 5 mL of 10% BSA to achieve a final concentration of 15 mM.Oleic Acid (OA): (1) Dissolve 50 mg of oleic acid in 0.8 mL of 0.1 M NaOH. (2) Combine 0.8 mL of the above solution with 10 mL of 10% BSA to achieve a final concentration of 15 mM.FFA Mixture: Prepare a mixture of FFA with a molar ratio of OA:PA = 2:1.

### 2.6. Oil Red O Staining for Cells

HepG2 cells were treated with 0.5 mM FFA only or FFA and 10 μM DAPA for 24 h, then washed 3 times with PBS, fixed with 4% formaldehyde for 20 min, washed 3 times with PBS, stained with Oil Red O (Nanjing Institute of Biotechnology) for 15 min, re-stained with nucleus for 1 min, and washed 3 times with PBS. The cells were observed and photographed under an electron microscope (Zeiss, Germany). ImageJ 1.8.0 software was used for Oil Red O quantification, specifically to determine the percentage of the total area of red fat droplets relative to the entire 40× image area.

### 2.7. Detection of Mito-Tracker, ROS, and MMP

HepG2 cells were treated with 0.5 mM FFA only or FFA and 10 μM DAPA for 24 h and then incubated in a humid incubator at 37 °C with 5% CO_2_ in a 50 nM Mito-Tracker Deepred (Beyotime Institute of Biotechnology, Nantong, China) solution for 30 min. Subsequently, Hoechst was used to stain the nucleus. After three rounds of PBS washing, the cell morphology was observed using a Zeiss microscope under 640 nm excitation. We evaluated the cellular ROS levels using a detection kit provided by the Beyotime Institute of Biotechnology (Nantong, China) according to the supplier’s protocol. Diacetyldichlorofluorescein (DCFH-DA) was diluted with serum-free culture medium at a ratio of 1:1000 to a final concentration of 5 µM. The cell culture medium was removed and 200 µL volume of diluted DCFH-DA was added. The HepG2 cells were treated with DCFH-DA at 37 °C for 30 min in the dark. After three rounds of PBS washing to thoroughly remove DCFH-DA that had not entered the cells, the cell fluorescence intensity was detected using a fluorescence microscope (Carl Zeiss, Jena, Germany) under 488 nm excitation, and MMP was detected using the JC-1 assay kit (Beyotime Institute of Biotechnology, Nantong, China) according to the supplier’s instructions. In short, JC-1 was incubated with HepG2 cells at 37 °C for 20 min. JC-1 accumulation was detected using fluorescence microscopy (Carl Zeiss, Jena, Germany) under 488 nm and 561 nm excitation. The fluorescence intensity was measured using Image J 1.8.0 software, which was the average fluorescence intensity of the captured field of view. The average fluorescence intensity was calculated as follows: (Mean) = the total fluorescence intensity of the area (IntDen)/the area of the region. It was standardized using the area of the region and repeated independently three times.

### 2.8. Mitochondrial Respiration Analysis

Mitochondrial respiration was measured using the Seahorse XF96 Analyzer (Agilent Seahorse Bioscience, Santa Clara, CA, USA) according to the manufacturer’s instructions. Briefly, the assay medium was prepared by supplementing Seahorse XF Base Medium (pH 7.4) with a specific combination of nutrient substrates, including 10 mM glucose, 1 mM pyruvate, and 2 mM L-glutamine. Cells were inoculated into Seahorse 96-well XF Cell Culture Microplates at a density of 20,000 cells per well and incubated at 37 °C for 24 h. Prior to running the XF detection, the Seahorse XF Sensor Cartridge was hydrated with hydration reagent and maintained in a CO_2_-free incubator at 37 °C for 24 h. On the day of analysis, the sterile water was replaced with Seahorse XF Calibrant solution. Mitochondrial respiration was measured using pre-prepared and optimized Seahorse-specific Mito Stress Test Kits (Agilent). Specifically, the concentrations of oligomycin, carbonyl cyanide 4-(trifluoromethoxy)phenylhydrazone (FCCP), and rotenone/antimycin A were set at 1.0 μM, 3.0 μM, and 1.0 μM, respectively. Seahorse Wave Controller Software version 2.6.3 (Agilent Seahorse Bioscience, Santa Clara, CA, USA) was used to operate and control the Seahorse XF96 Analyzer. After completing the measurements, the data were exported for further processing and analysis.

### 2.9. Quantitative Real-Time PCR (qRT-PCR)

After treatment with 0.5 mM FFA and 10 µM DAPA for 24 h, AG RNAex Pro reagent (Accurate Biotechnology, Changsha, China) was used to isolate RNA from the cells. RNA concentration was determined on a Nanodrop 2000C spectrophotometer (Nano Drop Technologies, Wilmingto, DE, USA). Samples with an OD260/OD280 ratio ranging between 1.8 and 2.0 were used for experiments. An Evo M-MLV RT Premix kit (Accurate Biotechnology) was used to reverse transcribe RNA to cDNA following the supplier’s guidelines. cDNA was used as a template for quantitative real-time PCR using the LightCycler 480 II system (Roche) and SYBR Green Premix Pro Taq HS qPCR kit (Accurate Biotechnology). During the amplification process, a melting curve was generated to verify the presence of erroneous amplification or primer–dimer products. Using β-actin as an internal reference, the relative mRNA expression level of the target gene was calculated using the (2^−ΔΔCT^) method. The primer sequence is as follows in Table 1.

### 2.10. Western Blotting

After treatment with 0.5 mM FFA only or FFA and 10 µM DAPA for 24 h, cell lysis buffer incorporating a cocktail of protease inhibitors was used to break the cells. A BCA protein concentration determination kit (# P0012S, Beyotime, Nantong, China) was used to estimate the protein concentration. After adjusting the protein concentration, samples were boiled at 100 °C for 8 min, and then 15 µL of each sample was transferred into 10% sodium dodecyl sulfate acrylamide gel. Protein molecules were separated at a constant voltage of 120 V and transferred onto a nylon membrane. After sealing with 5% skim milk, the membrane was incubated with primary antibodies against NLRP3 (1:1000) (Abcam, Cambridge, MA, USA), ASC (1:1000) (Abclonal, Wuhan, China), Caspase-1 (1:1000) (Abclonal, Wuhan, China), P20 (1:1000) (Cell Signaling Technology, Danvers, MA, USA), GSDMD (Abcam, Cambridge, MA, USA), and β-actin (1:5000) (Abcam, Cambridge, MA, USA) overnight at 4 °C. The membrane was washed three times with Tris-buffered saline containing 0.1% Tween-20 for 5 min, and then incubated with secondary antibody (peroxidase-conjugated goat anti-rabbit or -mouse IgG) at room temperature for 1 h. Afterwards, the membrane was reacted with an enhanced chemiluminescence solution (FdBio, Science) and imaged. Protein bands were quantified using Image J 2.0 software. The ratio of the intensity of the target protein band to the intensity of the β-actin band was calculated.

### 2.11. ELISA

After treatment with 0.5 mM FFA only or FFA and 10 µM DAPA for 24 h, supernatants from cell cultures were collected, and the concentrations of IL-1β and IL-18 were determined in accordance with the instructions of the manufacturers (R&D, eBioscience, San Diego, CA, USA). All samples were analyzed in duplicate. The concentration was calculated using a standard curve.

### 2.12. Statistical Analyses

GraphPad Prism 8 software (GraphPad Inc., La Jolla, CA, USA) was used for the statistical analyses. The results are expressed as mean ± SEM. We conducted a normality test on the data. A two-tailed unpaired Student’s *t*-test was used to compare data between two groups, while ANOVA and appropriate post hoc analyses (Tukey’s multiple comparison test) were employed for multiple groups. Statistical significance was defined by a *p* value < 0.05.

## 3. Results

### 3.1. DAPA Improved Glucose and Lipid Metabolism in Diabetic Mice

To investigate the impact of DAPA on glucose and lipid metabolism in vivo, we examined the glucose and lipid metabolism in the diabetic model of high-fat diet (HFD) + streptozotocin (STZ) and db/db mice. After 8 weeks of DAPA treatment, the body weights compared with the model (HFD + STZ) group were not significantly different (*p* > 0.05) (Figure 1A). The group of mice treated with DAPA showed a significant reduction in fasting blood glucose levels (*p* < 0.001) compared with the model (HFD + STZ) group (Figure 1B). Insulin levels in the model group were elevated compared to those in the normal group, and DAPA treatment increased the plasma insulin levels (*p* < 0.05) (Figure 1C). The OGTT and ITT of the model group showed a deterioration in glucose tolerance and insulin sensitivity, which was significantly improved by DAPA treatment (*p* < 0.001) (Figure 1D,F), along with a significantly decreased AUC (*p* < 0.05) (Figure 1E,G). Compared with the CTR group, the liver index (liver weight/body weight), triglycerides (TC), and cholesterol (TG) levels in the HFD + STZ group showed a significant increase. These experimental indicators in the DAPA treatment group were significantly reduced compared to the HFD + STZ group (*p* < 0.05) (Figure 1H,J). The extent of liver damage was assessed by estimating the levels of ALT and AST, which were significantly increased in the db/db group in comparison to the control. The DAPA interventions decreased the increased levels in comparison to the db/db group (Figure 1K,L). These results indicate that the dyslipidemia and the liver lipid metabolism and function in db/db can be alleviated by DAPA treatment.

Similarly, DAPA treatment resulted in an improvement in glucose and lipid metabolism in the db/db mouse model (Figure 2A–J). Therefore, these results collectively indicate that DAPA treatment can improve glucose and lipid metabolism both in the HFD + STZ and db/db diabetic mouse models.

### 3.2. DAPA Regulated Hepatic Lipid Accumulation In Vivo and In Vitro

Gross specimens of liver showing the liver of the CTR-group mice were dark red in color, with a smooth surface and soft texture. The appearance of the liver in the db/db group mice included diffuse swelling with dull and thick edges, a lighter or grayish yellow surface color, and a greasy feeling. The hardness was slightly higher than the livers of the CTR-group mice. The liver size of the DAPA group tended to be normal compared to the model group, with a softer texture and a darker red color. H&E staining showed that the ectopic deposition of lipid and a large number of lipid drops were observed in the sections of db/db diabetic mice, while the DAPA treatment group showed significantly reduced severe lipid droplets in the liver tissue of db/db mice. Further studies using Oil Red O staining showed that DAPA treatment significantly reduced lipid accumulation, inflammatory cell infiltration, and focal necrosis in liver tissue of the db/db mice. Lipid deposition was observed in the mouse liver slices of our experiment, and the DAPA treatment group significantly improved this phenomenon (Figure 3A,B).

After validating the phenotype of the mice, we also wanted to observe the effects of DAPA treatment on in vitro cells and establish a FFA-induced liver lipid deposition and functional impairment model in HepG2 cells. We evaluated the in vitro cellular efficacy of DAPA and selected 10 μM DAPA for cell therapy. Oil Red O staining showed that 0.5 mM FFA treatment effectively induced ectopic lipid deposition in HepG2 cells, while the DAPA treatment group showed significantly reduced ectopic lipid deposition (Figure 4A,B). Therefore, our research results indicate that DAPA significantly regulates lipid deposition in the liver both in vivo and in vitro.

### 3.3. DAPA Intervention Improved FFA-Induced Mitochondrial Oxidative Stress

To further examine the ultrastructural changes in the liver, we conducted electron microscopy. In the normal group, mitochondria exhibited a regular and full morphology with clearly defined cristae. In contrast, the db/db model group displayed variability in mitochondrial size and shape, with visible swelling. In severe cases, mitochondria became vacuolated, and their cristae appeared disorganized or were completely absent. Additionally, there was a significant accumulation of lipid droplets of varying sizes, which, in extreme instances, coalesced to form large vacuoles. In the treatment group, mitochondrial morphology showed marked improvement, tending towards a more regular appearance with clearer cristae, and there was a substantial reduction in the number of lipid droplets (Figure 4C).

Mitochondria are the energy source for liver cell activity and the main cellular source of ROS. Oxidative stress can damage mitochondria and impair antioxidant defense mechanisms, leading to increased ROS production and inducing liver cell dysfunction. In order to further explore the mechanism of DAPA intervention, we next conducted Mito-Tracker, ROS, and JC-1 staining experiments to investigate the potential mitochondrial-related mechanism of DAPA treatment in reducing lipid accumulation in liver cells under conditions of FFA. Mito-Tracker analysis of HepG2 cells showed that the normal control group maintained uniform distribution of mitochondria, while cells treated with FFA exhibited aggregated and fragmented mitochondria. DAPA treatment restored the dynamic balance between mitochondrial fusion and fragmentation (Figure 4D,E).

High lipid toxicity leads to the production of mitochondrial ROS. When MMP decreases, mitochondrial ATP synthesis through oxidative phosphorylation is inhibited, resulting in damage to liver function [19]. Therefore, in order to further explore the improvement of mitochondrial ROS after DAPA intervention, we measured the ROS levels in the HepG2 cell line. According to mitochondrial ROS staining, ROS levels were reduced in the DAPA treatment group (Figure 4F,G). Meanwhile, compared with the control group, the PA group showed an increase in ROS levels. The above experimental results indicate that DAPA intervention significantly reduces mitochondrial ROS levels, thereby protecting mitochondrial function. FFA led to a decrease in mitochondrial MMP and an inhibition of mitochondrial ATP synthesis through oxidative phosphorylation. Therefore, in order to further explore whether the problem with mitochondrial ATP synthesis after DAPA intervention was caused by the influence of mitochondrial MMP sources, we measured MMP levels in cell lines. According to mitochondrial JC-1 staining, MMP improved in the DAPA treatment group (Figure 4H,I). Meanwhile, compared with the control group, the MMP levels in the PA group increased. The above experimental results indicate that DAPA intervention improves mitochondrial MMP levels, thereby protecting mitochondrial function.

Based on the above experiments, we can conclude that DAPA treatment corrected the imbalance between mitochondrial fusion and division in liver cells, while reducing ROS production and MMP decline, improving liver function. In order to further investigate mitochondrial function, we used a Seahorse energy metabolism analyzer . The effect of DAPA on regulating mitochondrial respiratory capacity in HepG2 cells compared to exposure to FFA was studied. In this regard, oxygen consumption rate (OCR) was used as an indicator of mitochondrial respiration. We found that FFA treatment inhibited the maximal respiration and ATP production of HepG2 cells, and DAPA treatment improved this condition (*p* < 0.05) (Figure 4J,K). This represented the maximum respiratory rate that cells could achieve to cope with metabolic challenges, as well as the reduced ATP synthesis capacity of mitochondria in meeting cellular energy demands by FFA treatment and recovered through DAPA. It also suggested that FFA may lead to further disruption of mitochondrial metabolism, and significant changes in mitochondrial dysfunction seem to be attributed to the role of DAPA.

These results collectively indicate that DAPA intervention improves mitochondrial-related functions such as cellular respiration, oxidative stress, and ATP synthesis in liver cells, corresponding to the previously observed improvement in mouse liver function.

### 3.4. DAPA Alleviates Inflammation and Hepatic Lipid Accumulation Through the NLRP3-Caspase-1 Signaling Axis

There is a strong correlation between NLRP3 and the progression of fatty liver, and DAPA is a specific inhibitor of NLRP3. Therefore, we used PCR, WB, and ELISA to detect the expression levels of NLRP3 and its downstream signaling pathways. Following DAPA treatment, the expression of NLRP3, ASC, Caspase-1, GSDMD mRNA and protein levels were reduced compared to the FFA model (*p* < 0.0001) (Figure 5A–D,I–R), which confirmed that DAPA specifically targets NLRP3 and its downstream signaling pathways in FFA-induced HepG2 cells and in mouse liver tissue. Moreover, in the DAPA group, there was an improvement in the mRNA levels of these inflammatory factor indicators (*IL-1β*, *IL-18*, *IL-6* and *TNF-α*), alleviating inflammation and damage of the liver (Figure 5E–H). In addition, the reduction of IL-1β and IL-18 in the cell culture supernatant after DAPA treatment further confirms this (Figure 5S,T). The above results collectively indicate that DAPA improves liver inflammation, oxidative stress, and liver function through the NLRP3-Caspase-1 signaling axis.

## 4. Discussion

The liver is an important organ for human metabolism. Liver lipid deposition is the main factor of MAFLD in type 2 diabetes patients. The accumulation of lipids in hepatocytes will trigger inflammatory reactions and oxidative stress and accelerate liver injury in patients with type 2 diabetes [1]. In type 2 diabetes, the liver may also be damaged, leading to MAFLD, in which FFA is converted into synthetic triglycerides, leading to lipid accumulation in hepatocytes. This accumulation can lead to liver inflammation and oxidative stress, ultimately resulting in hepatitis, cirrhosis, and more severe liver damage [3].

Furthermore, the presence of MAFLD can exacerbate the pathological processes of T2DM, forming a vicious cycle. MAFLD and T2DM are closely related and mutually causal and influential, and can promote or alleviate disease progression through mechanisms such as insulin resistance, liver cytokines, inflammation, and cellular aging. However, the current clinical efficacy for treating the comorbidities of T2DM-associated MAFLD is suboptimal, making the exploration of potential therapeutic drugs an urgent priority [20,21].

Research on NLRP3 inflammasome inhibitors is burgeoning [22,23,24]. DAPA is a novel, specific NLRP3 inhibitor [25]. It has shown promising results by enhancing cardiac function in Phase Ib clinical trials and has been proven safe for patients with heart failure. Unexpectedly, this study found that DAPA treatment led to an average decrease of 10.5 mg/dL in fasting blood glucose in a small group (*p* = 0.021), and a more significant decrease of 32.5 mg/dL in fasting blood glucose in 54% of patients with type 2 diabetes (*p* = 0.029), suggesting its potential to lower blood glucose levels [26].

In addition, preclinical studies of DAPA in gout-related diseases have entered phase II–III clinical trials and have achieved positive results [11,12,27]. In this study, DAPA has been shown to improve glucose and lipid metabolism in HFD + STZ and db/db diabetes mouse models. In addition, we observed an increase in insulin levels in the HFD + STZ mouse model group compared to the normal group, indicating the presence of hyperinsulinemia. After DAPA treatment, plasma insulin levels further increased. In type 2 diabetes mice, plasma insulin levels reflect insulin resistance and insulin secretion.

Although DAPA treatment can alleviate insulin resistance and hyperinsulinemia to some extent, it may also impact insulin secretion from the pancreas, leading to a significant increase in insulin secretion overall. Consequently, after DAPA treatment, the plasma insulin levels in the mice were even higher than those in the model group. This finding suggests that DAPA may have a potential effect on insulin secretion from the pancreatic islets.

The early stage of type 2 diabetes mellitus (T2DM)-associated MAFLD is characterized by lipid degeneration, with lipid accumulation being a typical feature [28,29,30].

In our animal experiment on diabetic mice, H&E and Oil Red O staining showed hepatic steatosis and lipid accumulation induced by hyperlipidemia. DAPA treatment improved liver lipid accumulation caused by hyperlipidemia. Further research on the diabetes mouse model shows that the increase in ALT and AST levels indicates liver cell injury and is an important indicator of the severity of liver injury. The upregulation of low-density lipoprotein (LDL), cholesterol, and triglycerides caused by high-fat and other metabolic stresses was reversed by DAPA, suggesting that DAPA can mitigate liver damage induced by high-fat metabolic pressures.

Consistent with the in vivo results, further validation using cell lines showed that DAPA alleviated hepatic steatosis and lipid accumulation caused by high-lipid metabolic stress. Literature reports indicate that lipid metabolism dysfunction is often accompanied by reactive oxygen species (ROS) production, leading to oxidative damage [31,32,33]. Oxidative stress, caused by the dysregulation of ROS and antioxidant systems, is considered a biomarker for, and a major cause of the progression of, hepatic lipid accumulation [34,35,36]. In this study, high-fat metabolic stress led to an increase in ROS levels, while DAPA effectively alleviated ROS changes and improved the pathological process of liver lipid accumulation. Excessive ROS can disrupt liver lipid homeostasis, but DAPA has antioxidant activity, thereby protecting liver lipid metabolism. These results confirm that the hepatoprotective effect of DAPA on hepatic lipid accumulation is closely related to its antioxidant properties.

Mitochondria play a crucial role in regulating liver lipid metabolism and oxidative stress [37,38,39]. High lipid metabolic stress can trigger oxidative stress [40,41].

Mitochondria are the main source of ROS, which increase with the production of ROS, leading to structural changes and mitochondrial dysfunction [42,43]. Recent studies have shown that elevated levels of mitochondrial ROS can trigger NLRP3-dependent inflammation through danger signals under high-fat stress or changes in the liver microenvironment. In addition, mitochondria produce ATP through oxidative phosphorylation, providing energy for liver cell function [42]. Our results indicate that under high-fat pressure, mitochondrial ROS increase and the balance between mitochondrial fusion and fission is disrupted. DAPA significantly reduced mitochondrial oxidative stress and improved mitochondrial morphology. Mitochondrial membrane potential (MMP) has been used as an indicator for evaluating mitochondrial dysfunction. Previous studies have shown that the membrane potential of liver cells under high-lipid metabolic stress tends to decrease, consistent with our findings. DAPA effectively restored the MMP levels, suggesting that its hepatoprotective effect is at least partially attributed to the improvement of mitochondrial dysfunction.

In addition, mitochondrial oxidative stress is closely linked to NLRP3 inflammasomes. Elevated ROS activate NLRP3 inflammasome-mediated pyroptosis, which in turn damages mitochondria, creating a vicious cycle. This association has also been observed in liver cells. Under metabolic stress, mitochondrial ROS levels significantly increase, releasing danger signals that trigger NLRP3-dependent pro-inflammatory responses and further release of danger signals. This cycle exacerbates mitochondrial damage, leading to interruptions in energy supply and dynamic imbalances in mitochondrial energy processing, ultimately resulting in liver injury [43]. In our study, we found that DAPA can target the NLRP3 pathway and may improve liver injury by enhancing the role of mitochondria in liver lipid metabolism.

Cellular pyroptosis and the release of inflammatory cytokines are associated with the activation of inflammasomes. Increasing evidence suggests that the activation of NLRP3 inflammasomes promotes hepatic lipid deposition and the progression of inflammation [44]. In our study, a high-fat diet induced the activation of the NLRP3 inflammasome in hepatocytes, triggering cell pyroptosis, releasing inflammatory cytokines, and contributing to the pathogenesis of diabetes-related liver fat deposition. DAPA specifically targets the NLRP3-Caspase-1 signaling axis to alleviate cell pyroptosis and reduce the release of inflammatory factors. DAPA can prevent cell pyroptosis and reduce liver inflammation, oxidative stress, and lipid deposition by inhibiting NLRP3, thereby improving liver function in mice. Our results show for the first time that DAPA plays an important role in the treatment of diabetes-related liver fat deposition by targeting NLRP3-Caspase-1 axis-mediated cell pyroptosis.

## 5. Limitations

Our research findings suggest that targeting NLRP3 may provide a new strategy for the treatment of T2DM-related MAFLD. However, this study still has some limitations. Firstly, the precise binding site and specific mechanism of action between DAPA and NLRP3 targeting are still unclear, and further research and exploration are warranted. Secondly, due to the complexity of the liver microenvironment, current work mainly focuses on changes and mechanisms related to lipid deposition in liver cells. It is worth noting that the roles of other cell types in DAPA-mediated liver function protection still need to be elucidated. In addition, although the treatment duration in our animal model was relatively short, significant statistical differences were observed compared to the model group. The selection of male mice for this study was based on classic literature to avoid potential experimental bias introduced by estrogen. The main reasons for using male mice instead of female mice are as follows: (1) Hormonal effects: Female mice experience hormonal fluctuations during their reproductive cycle, which can cause changes in their behavior and physiological state, thereby affecting the stability and reproducibility of experimental results. (2) Behavioral differences: Female mice exhibit more anxious behavior during estrus, while male mice are relatively stable. This behavioral difference makes researchers more inclined to use male mice to reduce variables in experiments. (3) In addition, scientists have long believed that research on female animals is more complex because the effects of female hormones can complicate research questions. This concept leads researchers to prefer using male mice for experiments. Although HFD + STZ is a common method to induce diabetes in mice, it is also a mouse model of T2DM combined with NAFLD. However, db/db mice develop disease very rapidly, and the degree of abnormal glucose and lipid metabolism can be very severe [45].

## 6. Conclusions

This study demonstrated the potential use of the new NLRP3 inflammatory body inhibitor DAPA in the treatment of T2DM-related MAFLD, especially its role and mechanism in reducing liver lipid deposition and inflammation, protecting and restoring liver cell function. DAPA may improve liver glucose and lipid metabolism and function in diabetes model mice by inhibiting NLRP3-Caspase-1 signal transduction and regulating mitochondrial oxidative stress.

## Figures and Tables

**Figure 1 cimb-47-00148-f001:**
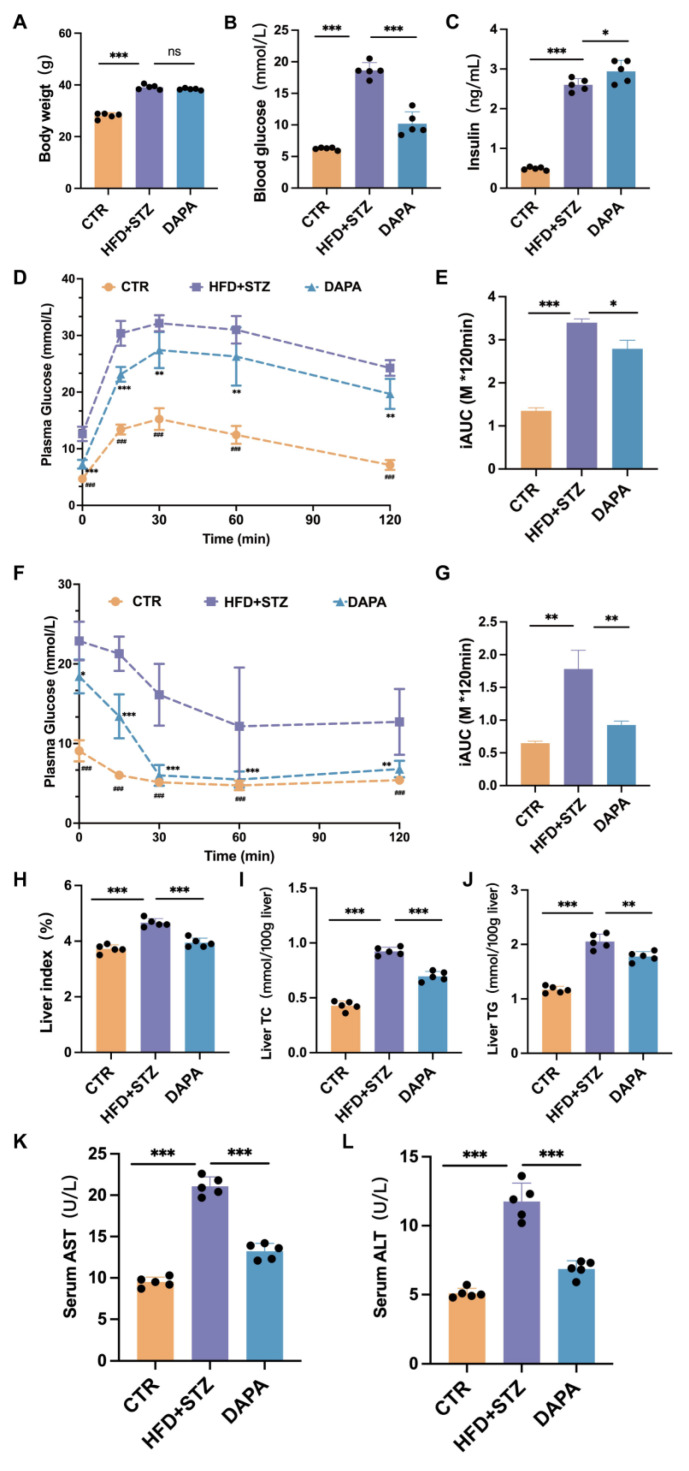
Dapansutrile improved glucose and lipid metabolism in the HFD + STZ mouse model. (**A**) Body weight monitoring in the HFD + STZ mouse model (*n* = 5 mice per group). (**B**) Plasma glucose levels in the HFD + STZ mouse model (*n* = 5 mice per group). (**C**) Insulin levels in the HFD + STZ mouse model (*n* = 5 mice per group). (**D**,**E**) OGTT and AUC in the HFD + STZ mouse model (*n* = 5 mice per group). (**F**,**G**) ITT and AUC of the HFD + STZ mouse model (*n* = 5 mice per group). (**H**) Liver index (liver weight/body weight). (**I**) Liver total cholesterol (*n* = 5). (**J**) Liver triglycerides (*n* = 5). (**K**) Serum AST (*n* = 5). (**L**) Serum ALT (*n* = 5). Data are expressed as means ± SEM and were analyzed using one-way or two-way ANOVA and appropriate post hoc analyses (Tukey’s multiple comparison test). * *p* < 0.05, ** *p* < 0.01, *** *p* < 0.001, ns means not significant. Abbreviations: CTR: control; DAPA: Dapansutrile; HFD: high-fat diet; STZ: Streptozotocin; OGTT: oral glucose tolerance test; AUC: area under the curve; ITT: insulin tolerance test.

**Figure 2 cimb-47-00148-f002:**
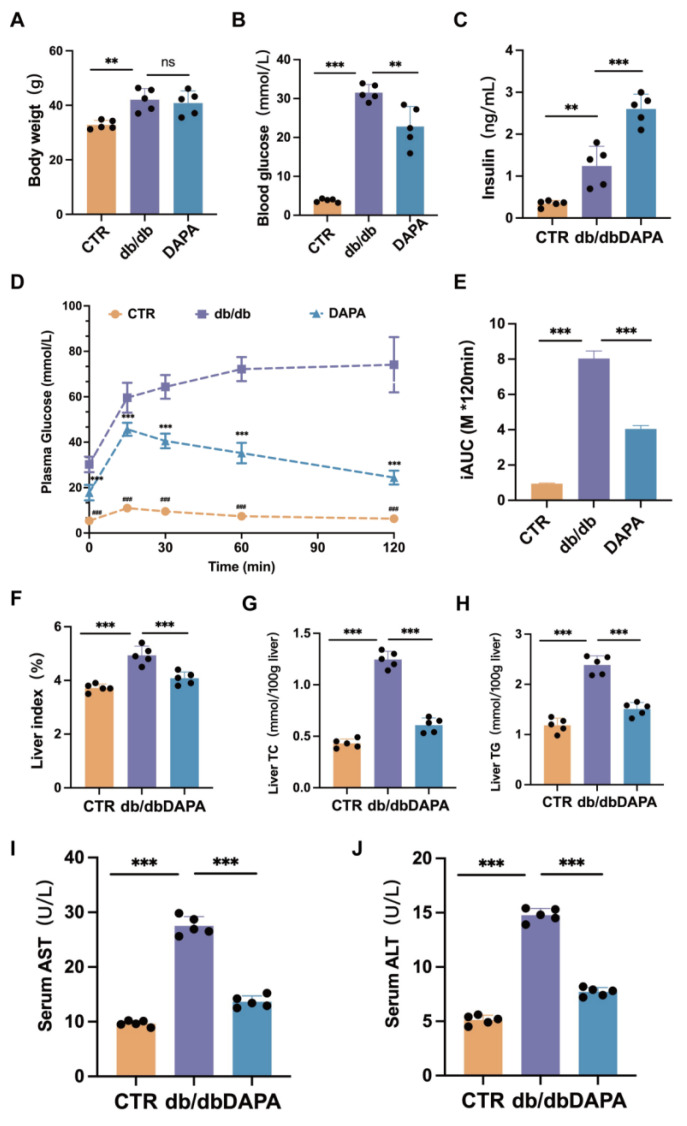
Dapansutrile improved glucose and lipid metabolism in the db/db mouse model. (**A**) Body weight monitoring in the db/db mouse model (*n* = 5 mice per group). (**B**) Plasma glucose levels in the db/db mouse model (*n* = 5 mice per group). (**C**) Insulin levels in the db/db mouse model (*n* = 5 mice per group). (**D**,**E**) OGTT and AUC in the db/db mouse model (*n* = 5 mice per group). (**F**) Liver index (liver weight/body weight). (**G**) Liver total cholesterol (*n* = 5). (**H**) Liver triglycerides (*n* = 5). (**I**) Serum AST (*n* = 5). (**J**) Serum ALT (*n* = 5). Data are expressed as means ± SEM and were analyzed using one-way or two-way ANOVA and appropriate post hoc analyses (Tukey’s multiple comparison test). ** *p* < 0.01, *** *p* < 0.001, ns means not significant. Abbreviations: CTR: control; DAPA: Dapansutrile; HFD: high-fat diet; STZ: Streptozotocin; OGTT: oral glucose tolerance test; AUC: area under the curve; ITT: insulin tolerance test.

**Figure 3 cimb-47-00148-f003:**
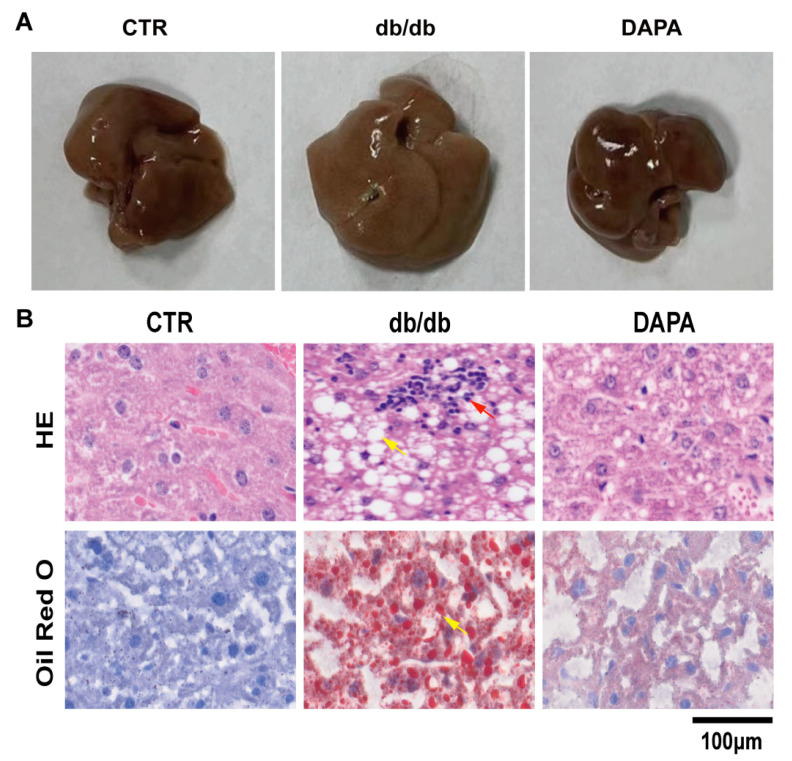
Dapansutrile alleviated liver lipid deposition. (**A**,**B**) Representative images of gross morphology, and liver tissues stained with H&E and Oil red O (*n* = 3). The yellow arrow indicates lipid droplets, and the red arrow indicates inflammatory cell infiltration and focal necrosis.

**Figure 4 cimb-47-00148-f004:**
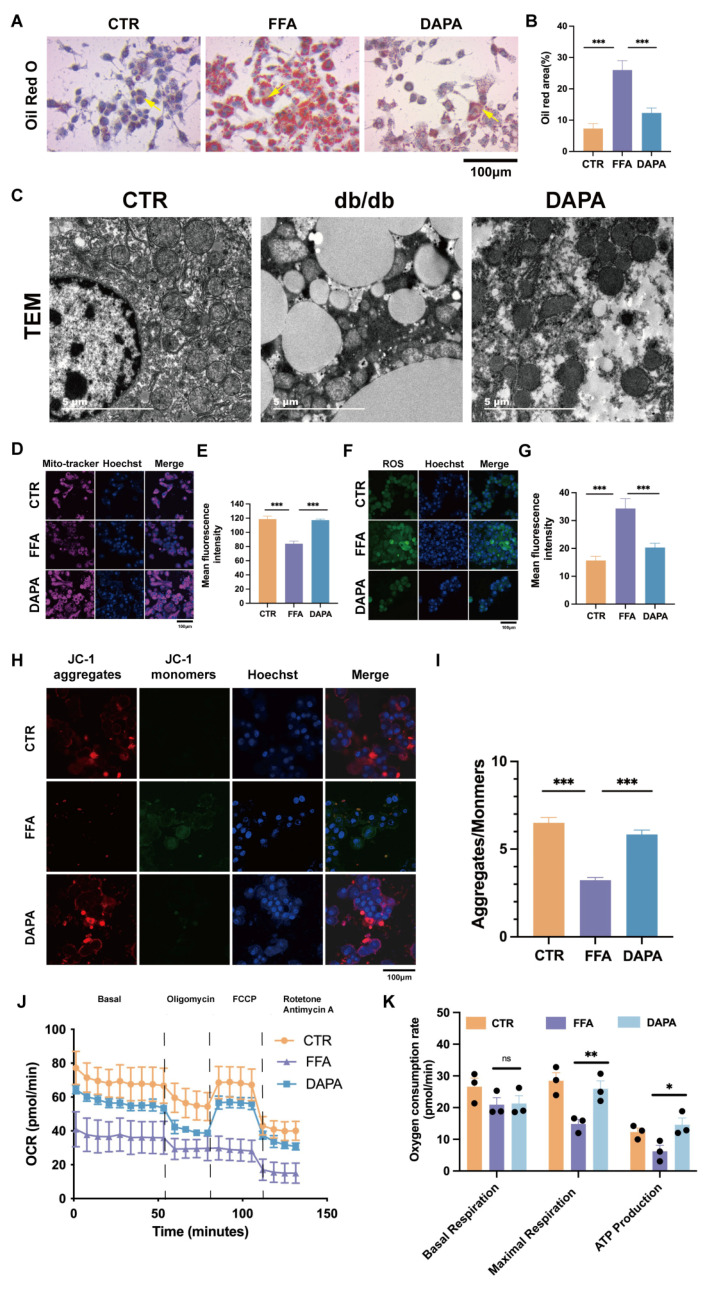
Dapansutrile improved lipid deposition in FFA-treated HepG2 cells by enhancing mitochondrial function. (**A**,**B**) Representative images of Oil Red O staining (*n* = 3). The yellow arrow indicates lipid droplets. (**C**) Representative images of TEM (**D**,**E**) Mito-Tracker Deep Red of HepG2 cells. (**F**,**G**) Measurements of mitochondrial ROS levels in HepG2 cells. (**H**,**I**) JC-1 staining of mitochondria in HepG2 cells. (**J**,**K**) Oxygen consumption rate (OCR) analysis of hepG2 cells (*n* = 3). CTR group: control; FFA group: 0.5 mM FFA; DAPA: 0.5 mM FFA + 10 μM DAPA. Data are expressed as means ± SEM and were analyzed using one-way or two-way ANOVA and appropriate post hoc analyses (Tukey’s multiple comparison test). * *p* < 0.05, ** *p* < 0.01, *** *p* < 0.001, ns means not significant. Abbreviations: CTR: control; FFA: free fatty acid; DAPA: Dapansutrile; NLRP3: NLR family pyrin domain-containing.

**Figure 5 cimb-47-00148-f005:**
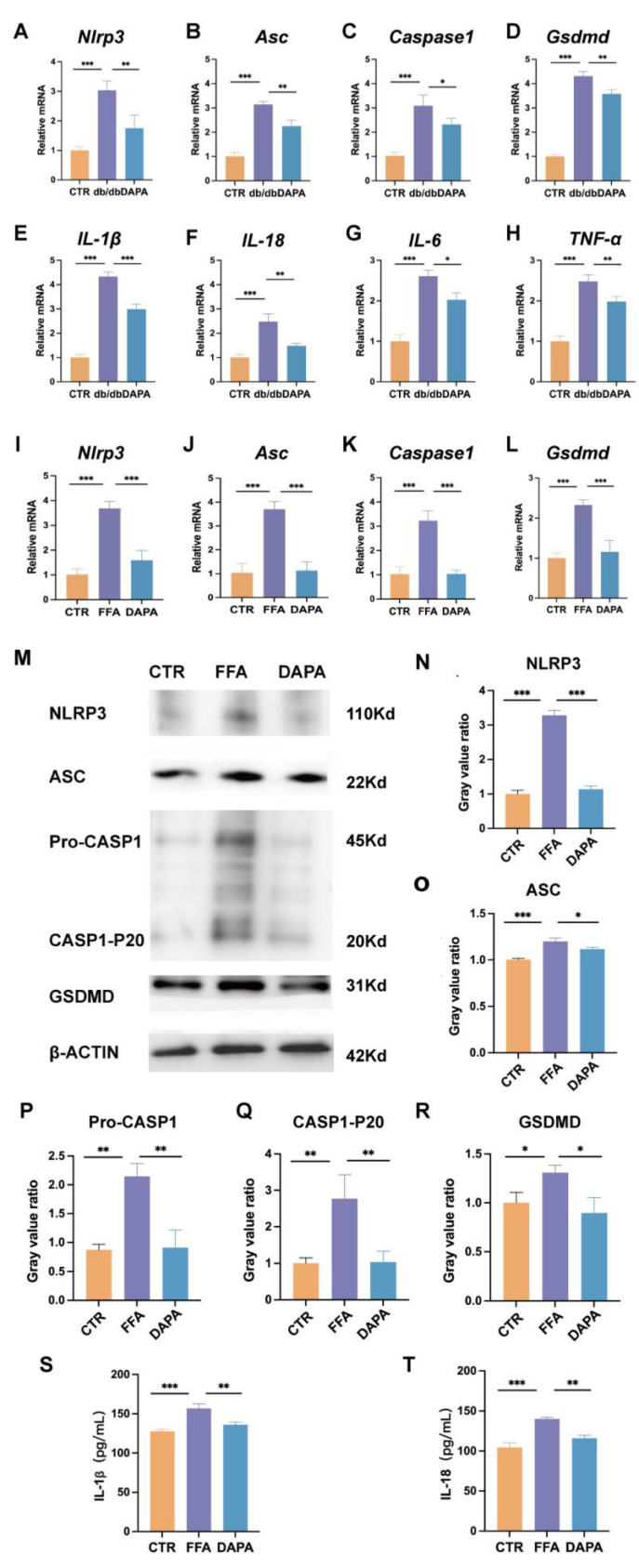
Dapansutrile alleviated liver cell damage in FFA-treated HepG2 cells by targeting the NLRP3-Caspase-1 signaling axis. (**A**–**H**) Gene expression in mouse liver tissue of *NLRP3* (**A**), *Asc* (**B**), *Caspase-1* (**C**), *Gsdmd* (**D**), *IL-1β* (**E**), *IL-18* (**F**), *IL-6* (**G**), and *TNF-α* (**H**). (**I**–**L**) Gene expression in HepG2 of *NLRP3* (**A**), *Asc* (**B**), *Caspase-1* (**C**) and *Gsdmd* (**D**). (**M**–**R**) Western blot of NLRP3, ASC, Pro-Caspase-1, p20, and GSDMD in HepG2. (**S**,**T**) IL-1β and IL-18 release of cell supernatants. CTR group: control; FFA group: 0.5 mM FFA; DAPA: 0.5 mM FFA + 10 μM DAPA. Data are expressed as means ± SEM and were analyzed using one-way ANOVA and appropriate post hoc analyses (Tukey’s multiple comparison test). * *p* < 0.05, ** *p* < 0.01, *** *p* < 0.001. Abbreviations: CTR: control; FFA: free fatty acid; DAPA: Dapansutrile; NLRP3: NLR family pyrin domain-containing 3; CASP1: Caspase-1.

**Table 1 cimb-47-00148-t001:** The sequences of RT-qPCR primers.

Target	Sequence	
q-RT-PCR (*Homo sapiens*)		
*β-actin*	F	5′-GCCTCGCCTTTGCCGAT-3′
	R	5′-AGGTAGTCAGTCAGGTCCCG-3′
*Nlrp3*	F	5′-AGGGATGAGAGTGTTGTGTGAAACG-3′
	R	5′-GCTTCTGGTTGCTGCTGAGGAC-3′
*Asc*	F	5′-AGCTCACCGCTAACGTGCTGC-3′
	R	5′-GCTTGGCTGCCGACTGAGGAG-3′
*Caspase-1*	F	5′-GCCTGTTCCTGTGATGTGGAG-3
	R	5′-TGCCCACAGACATTCATACAGTTTC-3′
*GSDMD*	F	5′-GCTGACCTCTGCCCTCCTTC-3′
	R	5′-TGGTGTGTGCGTTGGAATGC-3′
q-RT-PCR (*Mus musculus*)		
*Nlrp3*	F	5′-GAGTCTTCGCTGCTATGT-3′
	R	5′-ACCTTCACGTCTGGTTC-3′
*Asc*	F	5′-ACAATGACTGTGCTTAGAGAC-3′
	R	5′-CACAGCTCCAGACTCTTCTTTA-3′
*Caspase-1*	F	5′-GGCCCAGGAACAATGGCTGC-3′
	R	5′-GGGTCACAGCCAGTCCTCTTA-3′
*GSDMD*	F	5′-CTAGCTAAGGCTCTGGAGACAA-3′
	R	5′-GATTCTTTTTCATCCCAGCAGTC-3′
*IL1β*	F	5′-TGCCACCTTTTGACAGTGATG-3′
	R	5′-TGATACTGCCTGCCTGAAGC-3′
*IL18*	F	5′-AGACCTGGAATCAGACAACTTT-3′
	R	5′-TCAGTCATATCCTCGAACACAG-3′
*IL6*	F	5′-CTTCTTGGGACTGATGCTGGTGAC-3′
	R	5′-TCTGTTGGGAGTGGTATCCTCTGTG-3′
*TNFα*	F	5′-CGCTCTTCTGTCTACTGAACTTCGG-3′
	R	5′-GTGGTTTGTGAGTGTGAGGGTCTG-3′
*β-actin*	F	5′-GAGATTACTGCTCTGGCTCCTA-3ʹ
	R	5′-GGACTCATCGTACTCCTGCTTG-3′

## Data Availability

Data will be made available upon request.

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
