# Peer review of "Dapansutrile Regulates Mitochondrial Oxidative Stress and Reduces Hepatic Lipid Accumulation in Diabetic Mice"

_cimb, 2025, doi:10.3390/cimb47030148_

Round 1
Reviewer 1 Report
Comments and Suggestions for Authors
In this study, Wu and Zhou addressed the potential benefits of dapansutrile treatment in diverse diabetes study models including in vitro and in vivo ones. The authors performed several experiments including tolerance tests, biomarker assessment, hystology, qRT-PCR, Western blotting, ELISA and Seahorse metabolic analysis to pinpoint NLRP3 as an effective target for DAPA. While the paper is mostly well-written, this reviewer has several concerns that preclude further consideration of this manuscript:
Concerns
Sample Size Issues
Please include n values for liver sections on Figs. 3 and 4.
The Seahorse assay (n=3) might not be sufficient to draw firm conclusions about mitochondrial respiration (see below for more concerns on the respiration assay).
Lack of Proper Controls in In Vitro Studies
The HepG2 cell model is widely used yet it is a hepatocelular carcinoma cell line, which might not fully replicate normal hepatocyte metabolism.
It is unclear whether DAPA alone (without FFA) was tested to rule out off-target effects.
Mitochondrial Functional Analysis
Mitochondrial function was assessed primarily through OCR, MMP, and ROS assays. However, additional markers of mito function could provide a more complete picture (i.e. respiratory control ratios). In some cases, the “maximal respiration” looks lower than the basal respiration…do you have an explanation for that?
Inflammatory Markers and Insulin Signalling
While the study measures IL-1β and IL-18, other important inflammatory mediators (i.e., TNF-α, IL-6) could be further analyzed in serum and liver tissue to strengthen claims of inflammation modulation mediated by DAPA.
Please consider measuring Akt/ p-Akt levels .
Statistical Analysis
The normality of the data was tested, but details on whether variance was equal (i.e. Levene’s test) before applying t-tests or ANOVA are not provided in the manuscript.
Data Visualization
Some figure legends lack details on specific statistical tests used.
Minor
Please double check the list of accronyms used for missing information (i.e. FFA, etc).
Introduction Section
The transition from general MAFLD pathophysiology to DAPA’s potential therapeutic role is somewhat abrupt. Adding a brief discussion on why NLRP3 inhibition is an attractive target before introducing DAPA would improve flow.
Some sentences are somewhat long and complex, making it sometimes difficult to follow.
Discussion Section
The discusion jumps between in vivo and in vitro findings without clear transitions. Structuring it with subheadings (e.g., “DAPA Effects on Glucose and Lipid Metabolism,” “DAPA and Mitochondrial Function” would improve readability.
While the study acknowledges DAPA’s clinical trials for gout and cardiovascular diseases, it does not critically compare its mechanism of action to existing MAFLD therapies (e.g., GLP-1 receptor agonists, SGLT2 inhibitors).
Limitations Section
The study mentions the lack of female mice but does not discuss potential sex-dependent effects.
This study address short treatment with DAPA but future studies should be suggested to examine the long-term effects of this treatment.
Author Response
Dear editor and Reviewers,
Thank you for your help on our manuscript entitled “Dapansutrile regulates mitochondrial oxidative stress and reduces hepatic lipid accumulation in diabetic mice” (cimb-3484455). Sorry to bother you, we have answered the reviewer's relevant questions and made corresponding revisions to the article. Thank you again for your help. Please take the time to review our manuscript. If you have any questions, please contact us promptly.
Reviewer #1: Major concerns:
- Please include n values for liver sections on Figs. 3 and 4.
Response: Thank you for your review and pointing this out. We added this point in the manuscript (Revised main manuscripts Page:10, Line: 312, Page:13, Line: 445).
- The Seahorse assay (n=3) might not be sufficient to draw firm conclusions about mitochondrial respiration (see below for more concerns on the respiration assay).
Response: Thank you for your review and pointing this out. We use the hippocampus experiment combined with electron microscopy, mitoback, MMP, and ROS to illustrate the problem.
- The HepG2 cell model is widely used yet it is a hepatocelular carcinoma cell line, which might not fully replicate normal hepatocyte metabolism.
Response: Thank you for your suggestion. Through literature review, it was found that HepG2 cells can also be used as a cell model for fatty liver research due to their ability to induce intracellular accumulation of triglycerides. Therefore, many studies in this disease direction use the HepG2 cell model.
- It is unclear whether DAPA alone (without FFA) was tested to rule out off-target effects.
Response: Thank you for your suggestion. We point this out as shown in Figures of A-E. We found that in mice, there were no significant differences in blood glucose, insulin, and glucose tolerance between the group treated with DAPA alone and the control group. On the HepG2 cell line, WB showed no significant difference in the expression of NLRP3 with the addition of DAPA target alone compared to the control group.
(A) Blood glucose levels in the HFD+STZ mouse model. (B) Insulin levels in the HFD+STZ mouse model. (C-D) OGTT and AUC in the HFD+STZ mouse model. (E) Western blot of NLRP3 in HepG2.
- Mitochondrial function was assessed primarily through OCR, MMP, and ROS assays. However, additional markers of mito function could provide a more complete picture (i.e. respiratory control ratios). In some cases, the “maximal respiration” looks lower than the basal respiration…do you have an explanation for that?
Response: Thank you for your suggestion. The curve chart is correct, and the statistical bar chart is calculated automatically by the application software. However, the software defaults to a regular template with three points per stimulus for automatic calculation. But after I added the points for each stimulus, the software's automatic calculation error occurred. After calculating according to the formula, the current result is that the maximum respiration is greater than the baseline respiration (CTR and DAPA group) ( Main manuscripts Page:14, Fig4 J-K).
- While the study measures IL-1β and IL-18, other important inflammatory mediators (i.e., TNF-α, IL-6) could be further analyzed in serum and liver tissue to strengthen claims of inflammation modulation mediated by DAPA.
Response: Thank you for your suggestion. We point this out as shown in Figure A ( Main manuscripts Page:15, Fig5).
- Please consider measuring Akt/ p-Akt levels.
Response: Thank you for your review and pointing this out. We have already presented these results as shown.
- The normality of the data was tested, but details on whether variance was equal (i.e. Levene’s test) before applying t-tests or ANOVA are not provided in the manuscript.
Response: Thank you for your review and pointing this out. We have verified that the homogeneity of variance is homogeneous.
- Some figure legends lack details on specific statistical tests used.
Response: Thank you for your review and pointing this out. We have added this point in the manuscript ( Main manuscripts Page: 9, Line: 291-292 , Page: 10-11, Line: 307-309).

Reviewer 2 Report
Comments and Suggestions for Authors
Here are some essential suggestions to improve the quality of the manuscript.
- In Fig 3b, Authors are suggested to indicate the mentioned histopathological changes in the images through arrows. Similar description is needed in Fig 4, histological images.
- Also mention what kind of the changes author wants to show in Fig 3a.
- In the line, ‘Lipid deposition was observed in the mouse slices of our experiment, and the DAPA treatment group significantly improved this phenomenon (Figure 3A-C).” Author mentioned Figure 3A-C, what and where is the C part of Figure 3?
- Authors are suggested to improve the results section. They should explain how they interpreted the results. Like how they have evaluated the JC1 staining for MMP for aggregates and monomers and how the structural change indicate the mitochondrial membrane potential. This would help the readers to understand the technique and its interpretation.
Substantial part of the discussion text is either AI-written or AI-refined. Such text is mentioned below. Authors are suggested to rewrite this part.
- “The liver is a vital organ for human metabolism…………………. hepatitis, cirrhosis, and more severe liver damage [3]”
- “Additionally, the preclinical research of DAPA in the ……………………………. levels reflect both insulin resistance and insulin secretion.”
- “In our animal experiments on diabetic ……………………. important indicators of liver injury severity.”
- “In this study, high-fat metabolic stress …………………… thereby protecting liver lipid metabolism.”
- “Mitochondria are the primary source …………………… mitochondrial oxidative stress and improved mitochondrial morphology.”
- “This cycle exacerbates mitochondrial …………………. in liver lipid metabolism.”
- “DAPA specifically targets the ………………… NLRP3-Caspase-1 axis-mediated cell pyroptosis.”
Author Response
Reviewer #2: Major concerns:
- In Fig 3b, Authors are suggested to indicate the mentioned histopathological changes in the images through arrows. Similar description is needed in Fig 4, histological images.
Response: Thank you for your review and pointing this out. We have modified this point in the manuscript ( Main manuscripts Page:11, Line: 329-330; Page:14-15, Line: 420-421).
- Also mention what kind of the changes author wants to show in Fig 3a.
Response: Thank you for your review and pointing this out. We have modified this point in the manuscript ( Main manuscripts Page:11, Line: 314-320).
- In the line, ‘Lipid deposition was observed in the mouse slices of our experiment, and the DAPA treatment group significantly improved this phenomenon (Figure 3A-C).” Author mentioned Figure 3A-C, what and where is the C part of Figure 3?
Response: Thank you for your review and pointing this out. Due to adjusting the puzzle, Figure 3C has been moved to Figure 4 and was forgotten to be deleted.
- Authors are suggested to improve the results section. They should explain how they interpreted the results. Like how they have evaluated the JC1 staining for MMP for aggregates and monomers and how the structural change indicate the mitochondrial membrane potential. This would help the readers to understand the technique and its interpretation.
Response: Thank you for your review and pointing this out. We have modified this point in the manuscript.
- Substantial part of the discussion text is either AI-written or AI-refined. Such text is mentioned below. Authors are suggested to rewrite this part.
“The liver is a vital organ for human metabolism…………………. hepatitis, cirrhosis, and more severe liver damage [3]”
“Additionally, the preclinical research of DAPA in the ……………………………. levels reflect both insulin resistance and insulin secretion.”
“In our animal experiments on diabetic ……………………. important indicators of liver injury severity.”
“In this study, high-fat metabolic stress …………………… thereby protecting liver lipid metabolism.”
“Mitochondria are the primary source …………………… mitochondrial oxidative stress and improved mitochondrial morphology.”
“This cycle exacerbates mitochondrial …………………. in liver lipid metabolism.”
“DAPA specifically targets the ………………… NLRP3-Caspase-1 axis-mediated cell pyroptosis.”
Response: Thank you for your review and pointing this out. We have modified this point in the manuscript ( Main manuscripts Page:2, Line: 520-526, 539-544, 552-556, 564-567, 571-578, 587-591, 597-601).

Round 2
Reviewer 1 Report
Comments and Suggestions for Authors
The authors have satisfactorily addressed most concerns raised by the reviewer, except the one pertaining to the lack of female mice in their experiments. The authors mention that "there is no obvious dependence on the drug itself in experiments", yet they lack evidence to support such claim. Adding a few sentences on such limitations of this study would be helpful for the readership.
Author Response
Dapansutrile regulates mitochondrial oxidative stress and reduces hepatic lipid accumulation in diabetic mice
Dear editor and Reviewers,
Thank you for your help on our manuscript entitled “Dapansutrile regulates mitochondrial oxidative stress and reduces hepatic lipid accumulation in diabetic mice” (cimb-3484455). Sorry to bother you, we have answered the reviewer's relevant questions and made corresponding revisions to the article. Thank you again for your help. Please take the time to review our manuscript. If you have any questions, please contact us promptly.
Reviewer #1: Minor concerns:
- The authors have satisfactorily addressed most concerns raised by the reviewer, except the one pertaining to the lack of female mice in their experiments. The authors mention that "there is no obvious dependence on the drug itself in experiments", yet they lack evidence to support such claim. Adding a few sentences on such limitations of this study would be helpful for the readership.
Response: Thank you for your review and pointing this out. We added this point in the manuscript (Revised main manuscripts Page:19, Line: 613-621).
